# Matrix-Specific Effects on Caffeine and Chlorogenic Acid Complexation in a Novel Extract of Whole *Coffea arabica* Coffee Cherry by NMR Spectroscopy

**DOI:** 10.3390/molecules27227803

**Published:** 2022-11-12

**Authors:** Boris Nemzer, John Edwards, Diganta Kalita

**Affiliations:** 1VDF FutureCeuticals, Inc., Momence, IL 60954, USA; 2Department of Food Science and Human Nutrition, University of Illinois at Urbana-Champaign, Urbana, IL 61801, USA; 3Process NMR Associates, LLC, Poughkeepsie, NY 12603, USA

**Keywords:** caffeine, chlorogenic acid, caffeine-chlorogenic acid complex, whole coffee cherry, NMR spectroscopy

## Abstract

Coffee cherry is a rich source of caffeine and chlorogenic acids. In this study we investigate the structural analysis of caffeine-enriched whole coffee cherry extracts, CEWCCE by using ^1^H and ^13^C NMR spectroscopy. The changes in ^1^H chemical shift data in NMR spectra of CEWCCE compared to pure caffeine indicated the formation of complexes between caffeine and chlorogenic acids in aqueous solution. The effect of complexation on the peak position of caffeoylquinic acid and caffeine resonance with increasing addition of caffeine was investigated. 2D NOESY experiments show the presence of cross-peaks that are due to the proximity of chlorogenic acid and caffeine molecules in stable complexes in protic solvents. The quantification data of caffeine by ^1^H qNMR was found to be in close agreement with the data obtained by HPLC analysis.

## 1. Introduction

Coffee products have a long history of use in food and beverages, enjoyed for their stimulatory, sensory, and organoleptic attributes. During the recent explosion in consumption of coffee products, there has been a growing interest in the nutritional and functional attributes of coffee products and their chemical constituents. Two primary varieties of coffee cherries are *Coffea arabica* (Arabica) and *Coffea canephora* (Robusta). The general chemical constituents of coffee cherries are alkaloids, trigonelline, proteins, free amino acids, carbohydrates, lipids, phenolic acids, flavonoids, organic acids, and fixed oil [1,2]. Consumption of coffee products has been linked to various health benefits including neuro and cardiovascular protection, promotion of weight loss as well as possible anti-cancer and anti-aging activities [3,4,5].

Caffeine (1,3,7-trimethylxanthine), the main alkaloid component in coffee fruits, is reported to have various neuroprotective capabilities as well as well documented abilities to modulate alertness through stimulation of the central nervous system, blood circulation and respiration [6,7,8]. However, consumption of high doses of caffeine can lead to detrimental effects on human health [9] such as sleep disorders and mild addiction [10]. According to the US Food and Drug Administration, caffeine is generally safe in moderate amounts (<400 mg daily) [11]. Another major and potentially bioactive constituent of coffee cherry is chlorogenic acid. Health benefits including antioxidant activities, antidiabetic properties, hepatoprotective, hypoglycemic, and antiviral activities have been reported [12,13]. In one study Dellalibera et al. reported that consumption of chlorogenic acids-rich green coffee extracts induced weight loss and increased lean/fat ratios in overweight volunteers [14]. Another chlorogenic acids-rich extract of coffee cherry has previously been reported to have strong antioxidant capacity and to have ability to increase blood and exosomal levels of brain-derived neurotrophic factor (BDNF), an essential protein for neuronal and brain health [15]. In another investigation, a crude caffeine extract that contained 90–95% caffeine and 5–10% bioactive compounds increased glucose uptake into human skeletal muscle cells and exhibited antioxidant and anti-inflammatory activity [16]. Thus, caffeine and chlorogenic acids from whole coffee cherry (including the bean and the surrounding fruit) have potential for beneficial effects individually and, potentially, in combination. Due to the limitations related to caffeine dosing, there is a new interest in the development of a coffee extract with moderate caffeine content that also contains significant amounts of natural bioactive compounds including coffee chlorogenic acids. Further, the biological targets and degree of activity may be dependent upon (a) certain specific ratios of these two active phytochemicals and (b) their interactions with the surrounding extract matrix.

Investigations into the specific matrix effects have taken varied paths [14,15,16,17]. Several studies have shown that caffeine behaves as a good complexing agent with vitamins, nucleic acids, phenolic compounds, and other biomolecules that could affect biological properties such as bioavailability, cellular absorptivity, and biochemical activities [18,19,20]. Of particular interest is the exemplary interaction described by Gorter et al. who isolated caffeine chlorogenate complexes from coffee [21]. Since then, caffeine chlorogenate systems have received special interest both by their isolation and their structural elucidation in solution. Martin et al. demonstrated that caffeine formed complexes with phenolic substrates such as methyl gallate, m-nitrobenzoic acid and potassium chlorogenate by hydrogen bonds, a polar interaction, and coordination with metal ion [22]. Horman and Viani et al. reported that caffeine chlorogenate complex formed 1:1 type п-molecular complex by hydrophobic interactions [23]. Additionally, caffeine chlorogenate complexation was proposed to be a key factor for the compartmentation of caffeine in coffee plants and the relative distribution of chlorogenic acid throughout the plant [24]. In order to characterize the chemical composition of caffeine and chlorogenic acid in coffee cherries, their extracts and in coffee brews, analytical techniques such as HPLC (High Performance Liquid Chromatography, UPLC (Ultra Performance Liquid Chromatography), LCMS (Liquid Chromatography Mass Spectrometry), UV-Vis (Ultra-Violet Visible) spectrophotometry and NMR (Nuclear Magnetic Resonance) have been employed [25,26,27]. Of these, NMR is one of the most effective tools for detailed structural elucidation of complex organic molecules in solution. A comprehensive analysis with structural determination and intermolecular bonding interactions can be achieved by use of ^1^H-^13^C NOESY (Nuclear Overhauser Effect Spectroscopy) NMR. D’Amelio et al. demonstrated the formation of caffeine chlorogenate complex as well as self-association of caffeine and chlorogenic acid in aqueous solution by using high resolution NMR spectroscopy [28,29].

Previously, we analyzed the detailed phytochemical composition of a caffeine-rich, whole coffee fruit extract, CEWCCE, by HPLC and LCMS/MS systems and reported 73% caffeine and 6% chlorogenic acid [25,26]. We follow that analysis in this current study by carrying out a structural investigation of caffeine and chlorogenic acid and their complexing behavior in a unique commercial coffee fruit extract CEWCCE using NMR spectroscopy. Additionally, we have studied the molecular interaction of caffeine with chlorogenic acids in aqueous solution in order to better understand their complexation behavior.

## 2. Results and Discussion

### 2.1. ^1^H NMR Spectra of CEWCCE

The CEWCCE was isolated by water and ethanol extraction followed by multistep purification process. To investigate the nature of chemical environment of caffeine in CEWCCE in solution we have performed a qualitative analysis of caffeine and chlorogenic chemical shift changes by ^1^H NMR spectroscopy. Figure 1 presents two main structural components of chlorogenic acids and caffeine.

To determine whether the chemical shifts are the same among natural sources of caffeine or are altered by the presence of polyphenol content in the matrix, we analyzed CEWCCE (73% caffeine and 6% polyphenols) compared to GCBE (99% caffeine). Figure 2 shows the comparative ^1^H NMR spectra of CEWCCE and pure caffeine, GCBE (99% caffeine) obtained from VDF FutureCeuticals, Inc., (Momence, IL, USA). Earlier reports indicated the major peaks characteristic of caffeine in the ^1^H NMR spectrum are for the three methyl groups (H11″, H10″, H12″) with chemical shifts of 3.270, 3.442, and 3.900 ppm, respectively, and another peak 7.876 ppm for aromatic protons (H8″). One of the methyl signals (H12″) is a doublet and shifted considerably downfield due to its attachment and coupling to nitrogen (the other two methyl signals are singlets) [30]. In a similar manner, CEWCCE showed a characteristic ^1^H NMR spectrum assigned to three methyl groups and exhibited the chemicals shift of 3.256 (H11″); 3.426 (H10″); 3.887 (H12″) ppm 7.859 (H8″) ppm, respectively, for the caffeine molecule. However, the CEWCCE spectrum shows a significant difference in the position of the caffeine proton chemical shifts compared to highly purified GCBE caffeine solutions. These changes in the position of the caffeine peaks indicates the interaction of caffeine with neighboring molecular species such as chlorogenic acids or other molecules with electronic environments (aromatic and olefinic) that have an affinity for stacking with caffeine in close association.

One difficulty that can arise in such studies is that the shift of caffeine peaks can also be affected by caffeine-caffeine self-association, changes in pH of the sample matrix in solution, and the concentration distribution of the molecules in solution. To avoid some of these competing interactions we have attempted, whenever possible, to use similar concentrations and caffeine additions as well as by utilizing phosphate-buffered D_2_O when making up the NMR samples.

Following the observation that the caffeine proton chemical shift was unique between CEWCCE and GCBE, and in order to distinguish between the complexation behavior of caffeine with the chlorogenic acids already present in CEWCCE we also investigated and compared the NMR spectra from simple mixtures of synthetic caffeine and chlorogenic acids. Mixtures of synthetic caffeine and chlorogenic acid, as well GCBE, and a 40% CGA extract from coffee were formulated to mimic the caffeine: chlorogenic acid composition of CEWCCE. Figure 3 shows the NMR spectra of all these species. We observed that the chemical shifts for the protons both in the CEWCCE and also in the mixture of caffeine and chlorogenic acid extract were higher than the observed shifts for simple mixtures of synthetic caffeine and chlorogenic acids. These results are shown in Table 1. For all of the spectra, a separate, distinct signal was not observed for free caffeine compared to signals observed for complexed caffeine. This is due to the fact that the chemical shift position of the caffeine signal represents an equilibrium position that is an average of the positions of caffeine as they undergo exchange in the complex. In solution, the caffeine is constantly associating and disassociating from the various positions plausible for interaction. The weighted average position of the caffeine signals is affected by the form of the chlorogenic acid isomer, other aromatic species (such as trigonelline), and other caffeine molecules. Based on such chemical shift data D’Amelio et al. demonstrated the formation of caffeine-chlorogenic acid complex in aqueous solution. The interaction between caffeine and various chlorogenic acid isomers was characterized by monitoring chemical shift deviations upon addition of chlorogenic acid to a solution of pure caffeine in D_2_O. The association constant was reported as 30 ± 4 M^−1^ [27,28]. Some other NMR studies were carried out to investigate the structural analysis of coffee bean extract and caffeine complexes [31,32]. In other previous studies, changes of chemical shift demonstrated the formation of complexes between eosinY and caffeine where the chemical shifts of the methyl protons shifted from 3.33, 3.50, 3.94 to 3.19, 3.39, 3.77, respectively, as the caffeine evolved from pure to complexed caffeine [33]. Kan et al. demonstrated the interaction of caffeine and nucleic acids considering adenylyl-3′5′-adenosine, polyadenylic acid and ribo-(A-A-G-C-U-U)_2_ helix as models where the reported the chemical shift was about 0.13 ppm from a solution of caffeine/adenine with 0.05 ratio [34].

In the extracts and mixtures analyzed in this current investigation differences in chemical shifts as well as changes in the nature of doublet patterns of the peaks from spin-spin coupling are due to the presence of several isomers of chlorogenic acids as well as other ingredients such as sugars and organic acids that could complex with the caffeine molecules. This observation is important since it is indicative of a specific matrix and compositional effects that together are driving the complexation behavior of caffeine in CEWCCE that cannot be duplicated by simple mixtures of synthetic caffeine and chlorogenic acids.

### 2.2. ^1^H NMR Studies of Solutions of Caffeine and Chlorogenic Acids

To further investigate the phenomenon of complexation of caffeine and chlorogenic acid we performed qualitative ^1^H NMR spectra analyses by adding caffeine to solutions of chlorogenic acid (CQA) and its isomers dicaffeoylquinic acids (diCQA) from the solution as described in Section 3.4. The relative shifts of both caffeine and the caffeoyl components of CQA were observed after sequential additions of caffeine. The complexation shifts of all observable protons were recorded to observe if larger shifts were observed in some samples, indicating that certain regions of the molecules’ structures were interacting more strongly with each other and producing larger complexation shifts. Figure 4 shows the stacked plot comparison of the six pure CQA without addition of caffeine. Figure 5, Figure 6, Figure 7 and Figure 8 shows relative chemical shift of caffeoylquinic acids to which a weighed amount of caffeine was systematically added. Chemical shifts were noted compared to a caffeine standard solution. Figure 5 shows a stacked plot of one of these experiments (sequential caffeine addition to 3,4-dicaffeoylquinic acid (3,4-diCQA)) and Figure 6 shows a superimposed plot of these same data showing the complexation shifts observed when caffeine interact with 3,4-diCQA.

It has been observed that in the case of caffeine protons there is an initial up-field shift at the lower concentration of caffeine (~9 mM) and that relative complexation shift is reduced with 27 mM and 45 mM of caffeine in the solutions. This observed reduction of the complexation shift is indicative of a less tightly bonded complex structure that develops as higher molar ratios of caffeine are added. The initial caffeine addition causes a strong up-field shift as the caffeoyl aromatic sp2 electrons interact with the complexed caffeine aromatic electrons and increase the shielding of the caffeine proton environment. However, as more and more caffeine is added, caffeine molecules in the complexation “stack” are further distanced from the CQA, and these distanced caffeine molecules are not impacted by as much electron shielding from the CQA sp2 electrons; consequently, the exchange average chemical-shift position of the caffeine peaks moves downfield with further additions toward the expected position of “free (and self-associated)” caffeine.

Figure 7 show the observed complexation shifts relative to “free” caffeine with the sequential addition of caffeine. The values have not been normalized to consider the different relative concentrations of the CQA and caffeine in solution. Some interesting trends are observed in Figure 7. H8″ proton is most strongly affected by the addition of more caffeine compared to the methyl resonances while after addition of 54 mM of caffeine that trend only reverses in the case of cryptochlorogenic acid. It was observed that the relative shift gets smaller with greater caffeine addition. In contrast, Figure 8 shows the relative complexation shifts that were observed for the caffeoyl aromatic and alkene resonances of the various CQAs with sequential addition of caffeine. For these CQA resonances, the increasing addition of caffeine continues to cause an up-field shift as the electron density of the complexed caffeine molecule gradually increases the shielding of the caffeoyl aromatic protons. This increasing shift following caffeine addition is the opposite of what is observed in the shifts of caffeine signals under the same conditions.

Our previous LCMS study indicated the presence of 17 phenolic acids and derivatives including caffeic acid, ferulic acid, as well as 112 chlorogenic acids and derivatives in the CEWCCE [26]. Figure 9 shows a stacked plot of the ^1^H NMR spectra obtained on CEWCCE. Chlorogenic acids are clearly observed, with protons observed for the quinic-derived protons at 2.0–2.5 ppm, the caffeoyl alkene protons at 5.8–6.3, and 7.0–7.4 ppm and caffeoyl aromatic protons at 6.3–6.8 ppm [28]. This is consistent with the previously observed direction of complexation shifts of the caffeoyl aromatic and alkenic protons and the caffeine protons that was observed in the data from the sequential addition of caffeine to pure CQA. At higher caffeine concentrations the caffeine signals shift downfield towards the chemicals shift position of pure caffeine in solution, while the caffeoyl signals shift up-field. It is noted that free quinic acid is not observed in the CEWCCE extracts.

To further understanding of this complexation behavior, we also acquired ^13^C spectra of CEWCCE in D_2_O (Figure 10). Earlier studies on ^13^C NMR of chlorogenic acid indicated the chemicals shift for each carbon atom have distinct differences along with their varied multiplicities signals. The spectrum clearly shows the three large methyl carbons of caffeine in the 30–36 ppm region. The peak observed at 25 ppm is due to acetic acid that appears to be present in the sample. The remaining caffeine molecule signals are obvious from their intensity and full assignment has been made and presented in Figure 10. The CQA quinic CH_2_ carbons appear in the 36–45 ppm region, and it should be noted that each different CQA isomer yields a single peak in this area. The same can be said about all the CQA caffeoyl and quinic carbons regions (quinic carboxylic acids C7 (180–185 ppm), caffeoyl ester carbon C9′ (170–172 ppm), CQA quinic alcohol/ether carbons C1,3,4,5 (66–82 ppm), CQA-caffeoyl aromatic, alkene and phenolic carbons (146–150 ppm and 110–130 ppm).

We also carried out ^1^H-^13^C HSQC 2D-Spectra (heteronuclear single quantum correlation NMR spectroscopy) for the CEWCCE and the spectrum is shown in Figure 11 where the correlation between observed ^1^H signals can be utilized to assign the ^13^C NMR spectrum of the coffeeberry extracts.

To understand the nature and extent of binding strength of caffeine with chlorogenic acids we carried out ^1^H-^1^H 2D-NOESY experiments at various mixing times that allow internuclear interactions to be observed that provide evidence of complex formation. NOESY experiments are 2-dimensional. ^1^H NMR spectrum appears along a diagonal plot due to contributions of the magnetization of protons that have not been changed in solution, while cross peaks appear outside the diagonal at intersecting coordinates between signals of protons that are in close proximity to each other. Thus, cross peaks signals observed in a 2D NOESY spectrum are indicative of interaction of protons that may result from their close proximities. In a completely dissolved system with no complex formation molecules tumble past each other at rapid rates that do not allow dipole–dipole interactions of protons on different molecules. However, when a complex is present an orientation of two molecules becomes fixed and they move through the solution as a complexed pair. In this situation the two complexed molecules are relatively stationary with respect to each other and there is time for protons dipole–dipole interaction between protons in close-proximity. This interaction will lead to the observation of a cross-peak at the intersection of the 2D linear projection of the ^1^H NMR resonances of the interacting protons. The intensity of the cross-peak is proportional to the strength of the dipole–dipole interaction which is mediated by distance. Close protons yield larger cross-peaks. Hence, it is possible to determine internuclear distances from the intensity of the cross peaks. Figure 12 shows the results of such a NOESY experiment on the whole coffee fruit extract CEWCCE. The two cross peak circles outside the diagonal on the 2D plot are indicative of the interaction of the CH in caffeine with the aromatic protons of the chlorogenic acid isomers. The small-amplitude cross peak seems to indicate that the caffeine proton is interacting with a specific chlorogenic isomer in the complex. The lack of large intensity cross peaks may be due to the rate of exchange of caffeine entering and exiting the complex formation. Similar structural features of the complex of caffeine and chlorogenic acid have been demonstrated by others using 2D-^1^H-^1^H-NOESY spectroscopy by D’Amelio et al. [28].

Finally, we conducted quantitative ^1^H qNMR analysis of caffeine found in CEWCCE using NIST-traceable maleic acid or DMS internal standards. Under quantitative NMR experimental conditions, the internal standard qNMR method [35,36,37] allows the integrated signal intensities of an internal standard peak and that of an analyte of interest in the sample (in this case caffeine) to be ratioed on a molar basis. With a knowledge of the molecular weights, the weighed masses of standard and sample, the number of protons that the integrated peaks represent, and the purity of the standard the analyte purity can be determined as:Pa = Pis × (Ia × Nis × Ma × Wis/Iis × Na × Mis × Wa)(1)
where P is the purity (mass fraction), I is the integrated signal area, N is the number of ^1^H atoms contributing to integral area, M is the molecular weight, and W is the weight (typically measured to the nearest 0.01 mg). The subscripts “a” and “is” refer to the “analyte” and the “internal standard”, respectively.

Table 2 shows the caffeine and other ingredients concentrations obtained from ^1^H qNMR analysis of CEWCCE

## 3. Materials and Method

### 3.1. Materials

Caffeine enriched whole coffee cherry CEWCCE (commercially marketed under the trade name of “Coffeeberry^®®^ Energy 70%”), Coffeeberry^®®^ 40% chlorogenic acids extract and Green Coffee Bean Extract (GCBE) (99% Caffeine) were supplied by VDF FutureCeuticals, Inc. (Momence, IL, USA).

### 3.2. Chemicals

D_2_O (99.9%D); Cambridge Isotope Laboratories Inc, (Andover, MA, USA), USP reference standard (USP, North Bethesda, MD, USA).^1^H qNMR Internal Standards: TraceCERT^®®^ NMR Standards- Dimethylsulfone, calcium formate, caffeine, chlorogenic acids; Sigma-Aldrich, St Louis, MO, USA

CQA Standards: Cryptochlorogenic Acid (4-caffeoylquinic acid), neochlorogenic acid (5-caffeoylquinic acid), 3,5di-caffeoylquinic acid, (3,5-diCQA) 3,4-dicaffeoylquinic acid (3,4-diCQA), 4,5-dicaffeoylquinic acid (4,5-diCQA), were obtained from Cerilliant Corporation, Round Rock, TX, USA.

### 3.3. NMR Spectroscopy

^1^H, ^13^C, 2D-NOESY and 2D-gHSQC NMR experiments were performed on a Varian Mercury-MVX300 spectrometer operating at 299.99 MHz for ^1^H and 75.44 MHz for ^13^C, equipped with a Varian 5 mm ATB Probe SN 4131. Typical ^1^H NMR experiments were performed with a 20 s relaxation delay, a π/3 pulse tip angle (67°), an acquisition time of 4.2688 s collecting 64k points over a 14.992 kHz spectral window. 32 or 64 scans were signal averaged. ^13^C NMR data was acquired with a π/4 pulse tip angle (45°), with a 4 s relaxation delay, and acquisition time of 0.984 s collecting 24.6 k data points over a 25 kHz spectral window. 5,000–12,000 transients were signal averaged. 2D-NOESY experiment was acquired with a 150 ms mixing time. 2D-edited-gHSQC was acquired with 80 scans and 128 increments collecting 1024 ^1^H data points with an acquisition time of 0.3327 s and a relaxation delay of 2 s. Data was linear predicted out to an FT of 2k × 1k points.

NMR Data Processing Software: Mestrelab MNova Version 14.1.1-24571—12 February 2019.

### 3.4. Sample Preparation for NMR Spectra

All samples were prepared by dissolving carefully weighed amounts of sample into D_2_O into a 5 mm NMR tube. All weighing was performed on a Radwag Analytical balance capable of ±0.01 mg—Model AS 82/200.X2 SN 586338, CT, USA. The samples of mixture of caffeine and chlorogenic acids were prepared by mixing the weight amounts of synthetic caffeine and chlorogenic acids synthetic as well as coffee cherry extracts maintaining the ratio of caffeine and chlorogenic acids as 10:1 to mimic the composition of CEWCCE. The solutions of each CQA isomers were prepared by dissolving in 650 µL of D_2_O resulting the final concentration 3,4 diCQA (3.4 mM), 3,5-diCQA (4.9 mM), 4,5-diCQA (3.3 mM), 4-CQA (9.8 mM) and 5-CQA (5.8 mM). The ^1^H NMR spectrum of each was recorded after addition of caffeine with final concentration 9, 27 and 45 mM.

## 4. Conclusions

The chemical shift data from ^1^H NMR of caffeine enriched whole coffee cherry extracts that contain other bioactive compounds showed evidence of complexation between caffeine and chlorogenic acids in aqueous solution. The complexation behavior of caffeine with the naturally occurring chlorogenic acids already present in CEWCCE was decidedly different from complexation observed by the simple physical mixtures of synthetic caffeine and chlorogenic acids and also from analogous mixtures of GCBE caffeine and CGA extracts from coffee fruit. 2D NOESY experiments show the presence of cross-peaks in CEWCCE that are due to the close-proximity of chlorogenic acid and caffeine molecules in stable complexes in protic solvents; the scale of which indicates the likelihood of caffeine interaction with specific chlorogenic isomers. Caffeine complexation characteristics, therefore, are highly dependent upon the composition of the entire matrix within which they are found. The quantification of caffeine and chlorogenic acid are in complete agreement with the HPLC results. Several studies reported that caffeine interacts with various biomolecules including nucleic acids, polyphenols and various drug molecules that affects their physiological properties including solubility, bioavailability, and pharmacological properties. However, based on our observations, these interactions and subsequent impacts to physiological properties are likely composition- and concentration-dependent. Having analyzed the detailed chemical composition and caffeine complexation behavior of CEWCCE, further studies should attempt to elucidate how these unique characteristics may translate to modulations of bioavailability and bioactivity. Moreover, caffeine enriched coffee extract (CEWCCE) wherein caffeine is complexed with bioactive coffee compounds may provide a greater potential for unique health benefits beyond those available from caffeine alone.

## Figures and Tables

**Figure 1 molecules-27-07803-f001:**
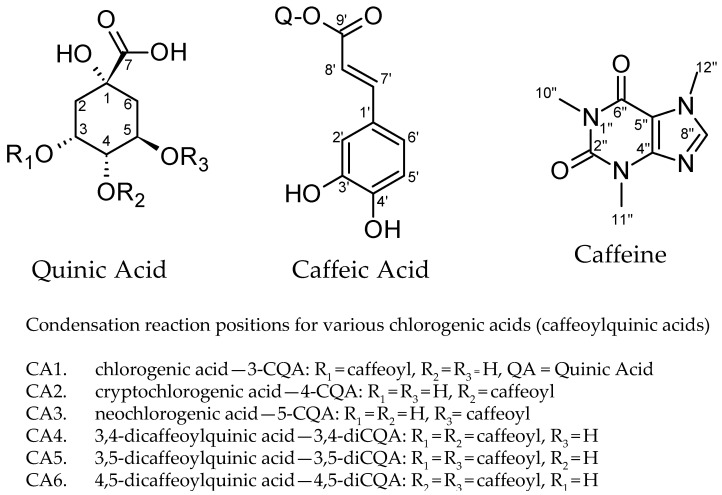
Carbon and proton positions utilized in the NMR spectrum annotations showing the chlorogenic acids utilized in this study of individual chlorogenic acid interactions as well as complex mixture interactions with caffeine.

**Figure 2 molecules-27-07803-f002:**
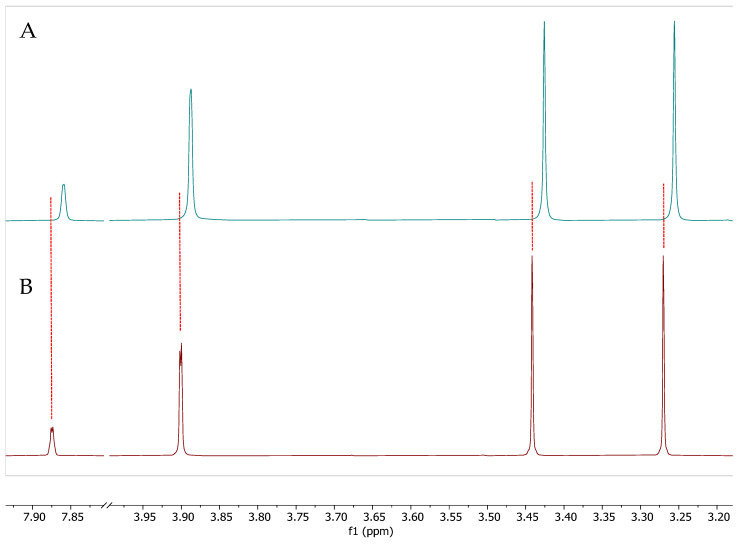
^1^H NMR spectra of (**A**) CEWCCE compared to the spectrum and (**B**) GCBE (99% Caffeine) in phosphate buffered D_2_O pD 7.1 operating at 299.99 MHz (Red: GCBE, Blue: CEWCCE). Dotted lines added for clarity of relative complexation shifts. Three methyl groups (H11″, H10″, H12″) with chemical shifts of 3.270, 3.442, and 3.900 ppm and another peak 7.876 ppm for aromatic protons (H8″).

**Figure 3 molecules-27-07803-f003:**
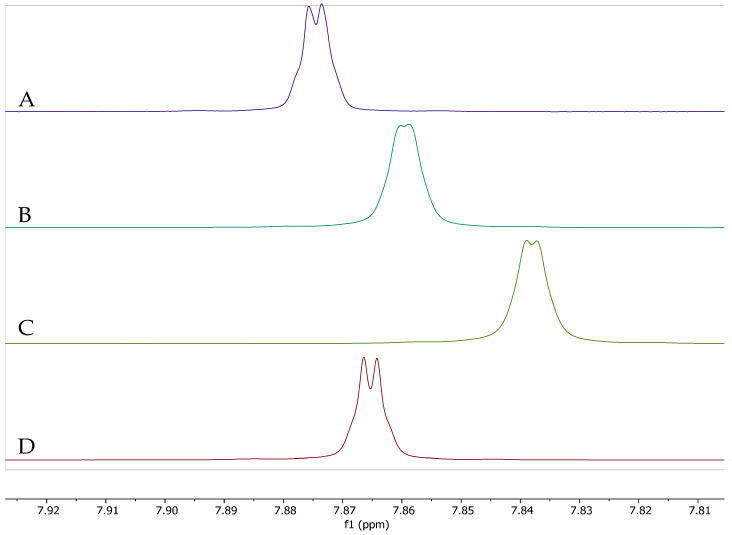
Aromatic caffeine signal H8 observed in the ^1^H NMR spectra of (**A**) GCBE (99% Caffeine) in phosphate buffered D_2_O pD 7.1 compared to (**B**) CEWCCE, (**C**) Mixtures of synthetic caffeine (Sigma, St Lois, MO, USA) and coffee cherry chlorogenic acids extracts containing 40% chlorogenic acids (obtained from VDF-FutureCeuticals), and (**D**) Mixture of synthetic caffeine and chlorogenic acids commercially available from Sigma.

**Figure 4 molecules-27-07803-f004:**
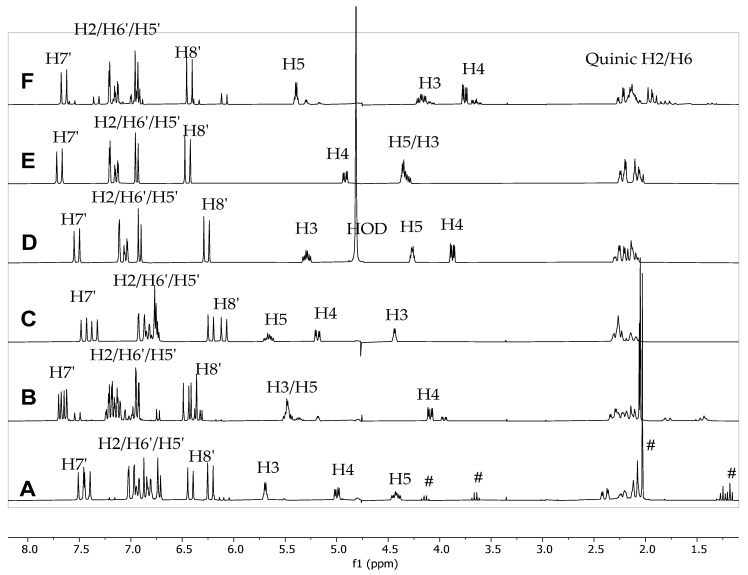
^1^H NMR stacked plot comparison of six chlorogenic acids investigated in this study without the addition of caffeine (**A**) 3,4-diCQA, (**B**) 3,5-diCQA, (**C**) 4,5-diCQA, (**D**) 3-CQA, (**E**) 4-CQA and (**F**) 5-CQA. # indicates ethanol and ethyl acetate residual solvent.

**Figure 5 molecules-27-07803-f005:**
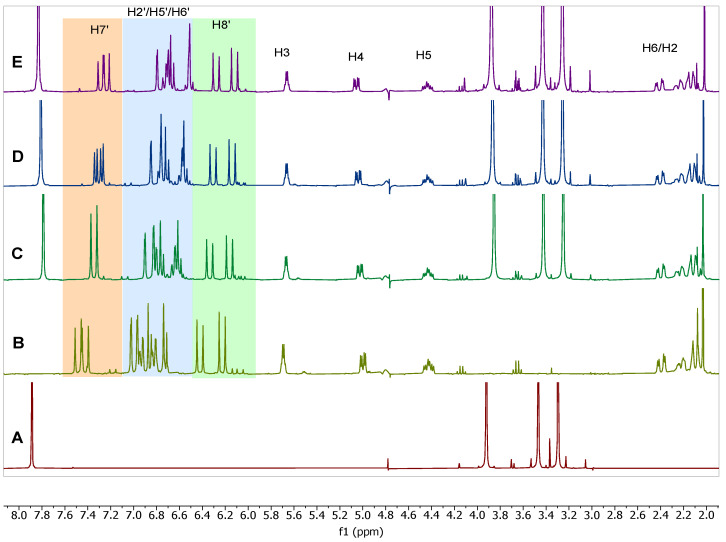
Effect of sequential caffeine addition to the ^1^H NMR chemical shifts of 3,4-diCQA (3.4 mM) and caffeine showing (**A**) pure caffeine, (**B**) 0 mM caffeine addition, (**C**) 9 mM caffeine addition, (**D**) 27 mM caffeine addition, (**E**) 45 mM caffeine addition.

**Figure 6 molecules-27-07803-f006:**
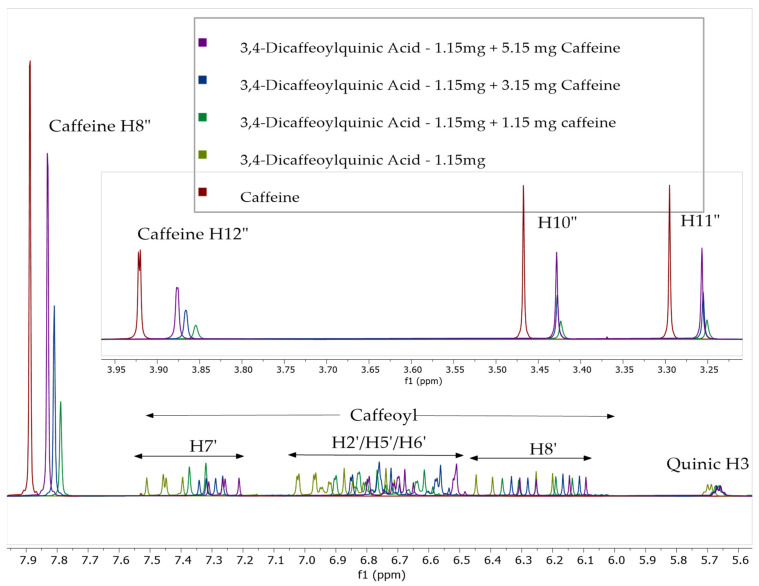
Superimposed ^1^H NMR spectra of 3,4-diCQA (3.4 mM) with sequential addition of caffeine (0 mM, 9 mM, 27 mM, and 45 mM).

**Figure 7 molecules-27-07803-f007:**
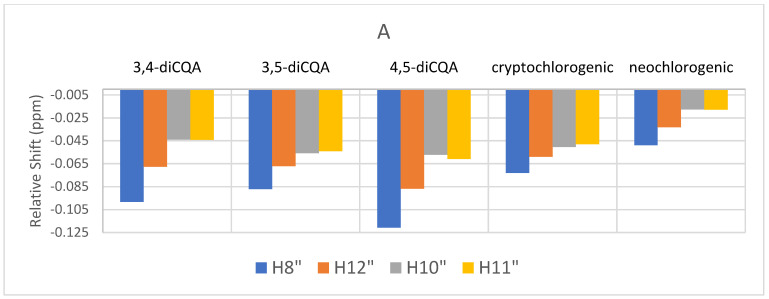
Relative complexation shifts of caffeine ^1^H NMR signals with sequential addition of pure caffeine to CQA isomers: (**A**) caffeine (9 mM) and chlorogenic acids (3–10 mM) (**B**) caffeine (27 mM) and chlorogenic acids (3–10 mM) (**C**) caffeine (45 mM) and chlorogenic acids (3–10 mM).

**Figure 8 molecules-27-07803-f008:**
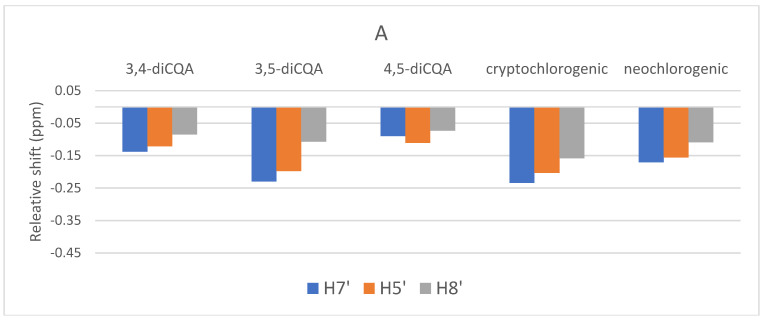
Relative complexation shifts of CQA caffeoyl ^1^H NMR signals with sequential addition of pure caffeine to CQA solutions; 3,4-diCQA (3.4 mM), 3,5-diCQA (4.9 mM), 4,5-diCQA (3.3 mM), crypto chlorogenic acid, 4-CQA (9.8 mM) and neochlorogenic acid, 5-CQA (5.8 mM). (**A**) 9 mM caffeine (**B**) 27 mM caffeine, and (**C**) 54 mM caffeine.

**Figure 9 molecules-27-07803-f009:**
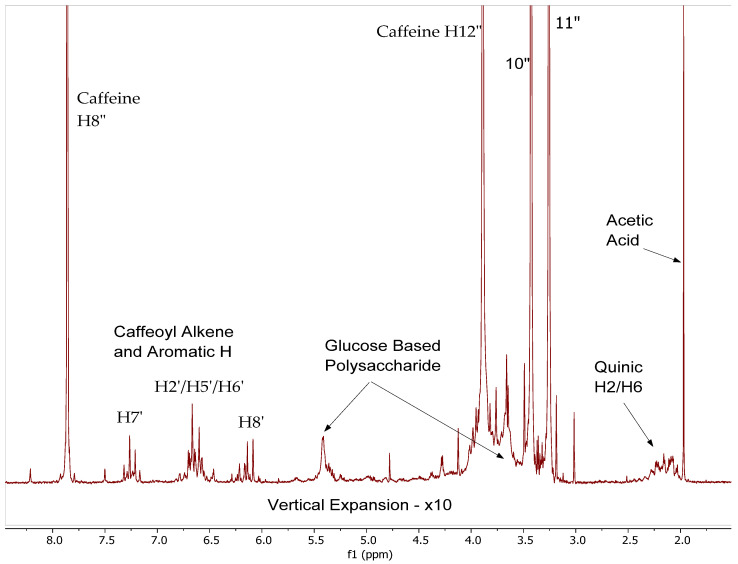
Caffeoyl region of the ^1^H NMR spectra of CEWCCE in D_2_O showing relative shifts in caffeine (H8″, 12″,10″,11″) and caffeoyl (H7′,2′,5′,6′,8′) resonances. Caffeine resonances are off scale to observe CQA signals.

**Figure 10 molecules-27-07803-f010:**
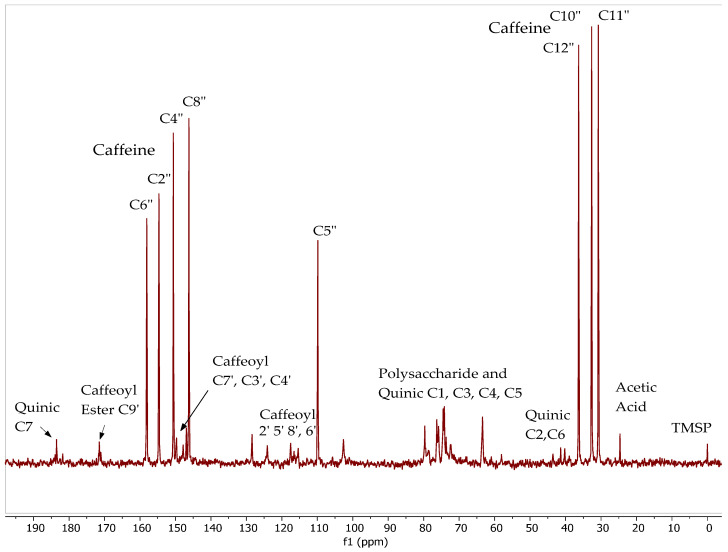
^13^C NMR spectra of CEWCCE in D_2_O operating at 75.44 MHz.

**Figure 11 molecules-27-07803-f011:**
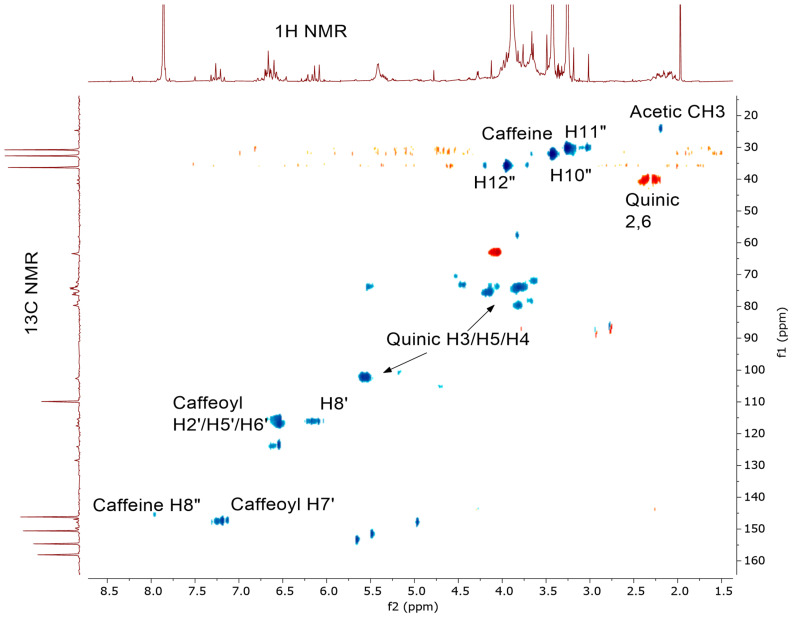
^1^H-^13^C gHSQC 2D-spectrum of CEWCCE in D_2_O.

**Figure 12 molecules-27-07803-f012:**
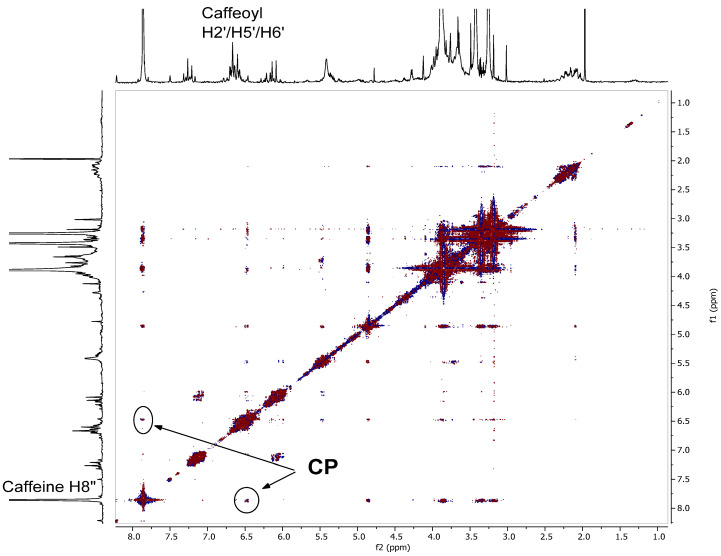
^1^H-^1^H 2D-NOESY NMR experiment with a mixing time of 150 ms performed on CEWCCE. Cross peaks indicated with “CP” demonstrate through-space close proximity of caffeine H8″ to the caffeoyl H2′/H5′/H6′ aromatic protons. Shows aromatic stacking in complex formation.

**Table 1 molecules-27-07803-t001:** Chemical shift data of ^1^H NMR spectra of caffeine, CEWCCE, and mixtures of caffeine and chlorogenic acids in phosphate-buffered D_2_O.

Sample	Chemical Shift, ppm (Carbon Number and Type)			
	H8 (CH)	H12” (CH3)	H10” (CH3)	H11” (CH3)
1. GCBE (Caffeine, 99%)	7.875	3.900	3.441	3.270
2. CEWCCE (70% Caffeine, 6% CA)	7.859	3.887	3.426	3.256
Chemical shift difference between 1 and 2	0.016	0.013	0.015	0.014
3. Mixture of Caffeine and CQA (natural extract)	7.838	3.831	3.329	3.163
Chemical shift difference between 1 and 3	0.037	0.069	0.112	0.107
4. Mixture of Caffeine and 5CQA (sigma)	7.865	3.850	3.340	3.173
Chemical shift difference between 1 and 4	0.010	0.050	0.101	0.097

**Table 2 molecules-27-07803-t002:** ^1^H qNMR–compositional analysis of CEWCCE.

CEWCCE	Composition (%)
Caffeine	73.25
Chlorogenic acid	7.63
Glucose	15.08
Malic acid	0.22
Acetic acid	0.47
Lactic acid	0.04
Ethanol	0.05

## Data Availability

Not applicable.

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
