# Peer review of "Matrix-Specific Effects on Caffeine and Chlorogenic Acid Complexation in a Novel Extract of Whole Coffea arabica Coffee Cherry by NMR Spectroscopy"

_molecules, 2022, doi:10.3390/molecules27227803_

Round 1

Reviewer 1 Report

My attention is actually drawn to one issue, which is the interaction between molecules. I would like the authors to write clearly what the stable complex between caffeine and chlorogenic acid looks like, how the molecules are positioned relative to each other, what exactly these interactions are. In the introduction, we can learn about hydrogen bonds, or pi-complex identified in previous studies. It is generally described in manuscript that there is a dipole-dipole interaction between protons in close proximity for the complex. The conclusions say that "2D NOESY experiments show the presence of cross-peaks in CEWCCE that are due to the close-proximity of chlorogenic acid and caffeine molecules in stable complexes in protic solvents". Here, a schematic drawing would turn out to be valuable to show or propose these specific interactions.

I would also like to point out that in Tab. 2 the word "Sample" is misleading, because the results rather show "sample content". The description of samples in Fig. 6 is illegible, and so is the font in Figures 11 and 12. In some places the font should be corrected.

Author Response

My attention is actually drawn to one issue, which is the interaction between molecules. I would like the authors to write clearly what the stable complex between caffeine and chlorogenic acid looks like, how the molecules are positioned relative to each other, what exactly these interactions are. In the introduction, we can learn about hydrogen bonds, or pi-complex identified in previous studies. It is generally described in manuscript that there is a dipole-dipole interaction between protons in close proximity for the complex. The conclusions say that "2D NOESY experiments show the presence of cross-peaks in CEWCCE that are due to the close-proximity of chlorogenic acid and caffeine molecules in stable complexes in protic solvents". Here, a schematic drawing would turn out to be valuable to show or propose these specific interactions.

  • We have investigated the possible interaction caffeine and chlorogenic acids for complexation in solutions through the NMR spectroscopy. Further studies with other spectroscopic and analytical techniques such as X-ray Crystallography and computational chemistry are expected to be useful to propose a structure of the complex. Therefore, we did not propose any structure from this study.

I would also like to point out that in Tab. 2 the word "Sample" is misleading, because the results rather show "sample content". The description of samples in Fig. 6 is illegible, and so is the font in Figures 11 and 12. In some places the font should be corrected.

  • We have updated these

Reviewer 2 Report

In this study, authors have explored the complexation of chlorogenic acid and caffeine by using 1H and 13C NMR spectroscopy, as well as their quantification, in coffee cherry. These results are of great relevance since coffee bioactive compounds play an important role for society, due to its high consumption. I recommend the publication in this journal. However, some details are missing, and some minor changes should be done:

1.      Did authors measure replicates for each sample? If so, provide confidence interval (for example: Table 2)

2.      Please, according to IUPAC recommendation, separate numbers from units where applicable (for example: 27 mM [page 8, line 209], 45mM [page 9, line 238], etc…)

3.      Some figures contain small fonts size inside them, please, provide replacement with highest fonts size (figure 6, figure 10, figure 11, figure 12).

Author Response

In this study, authors have explored the complexation of chlorogenic acid and caffeine by using 1H and 13C NMR spectroscopy, as well as their quantification, in coffee cherry. These results are of great relevance since coffee bioactive compounds play an important role for society, due to its high consumption. I recommend the publication in this journal. However, some details are missing, and some minor changes should be done:

  1. Did authors measure replicates for each sample? If so, provide confidence interval (for example: Table 2)
  • We did not run any replication. This study was investigated to verify with the available data obtained by HPLC.
  1. Please, according to IUPAC recommendation, separate numbers from units where applicable (for example: 27 mM [page 8, line 209], 45mM [page 9, line 238], etc…)
  • We have updated this
  1. Some figures contain small fonts size inside them, please, provide replacement with highest fonts size (figure 6, figure 10, figure 11, figure 12).
  • We have updated these.

Reviewer 3 Report

In this work, the authors have carried out the structural analysis of caffeine-enriched whole coffee cherry extracts, CEWCCE by using 1H and 13C NMR spectroscopy. The changes in 1H chemical shift data in NMR spectra of CEWCCE compared to pure caffeine indicated the formation of complexes between caffeine and chlorogenic acids 13 in aqueous solution. It is based on a rather classical methodology for NMR work The experimental and discussion sections were carried out carefully. For instance, the changes in proton chemical shifts are carefully interpreted taking into account other reasons that can lead to such changes. Therefore I consider it is valuable for publication in Molecules. However the following addressed points should be answered or completed:

 General remarks: 1) Indicate the proton frequency on the figures with proton NMR data (idem for 13C). 2) I not sure that four decimals are required for proton chemical shifts

 In Figure 1 the meaning of Q should be recalled

 Line 112 it is written “signals (H12’’) is a doublet” the origin of this pattern should be given.

 In Figure 2, to help the reading, the assignments of the NMR signals should be indicated on the different spectra

 In the caption of Figure 3 please insert H8''  after “Aromatic caffeine signal”

 In Figure 6, please try to improve the inset: font should be bigger and the black contour may be removed

 In Figure 9, font used for the NMR assignments should be bigger (same remarks for Figure 10)

 In Figure 12, the inset should be replaced by two arrows and the corresponding explanation should be inserted in the figure caption.

Author Response

In this work, the authors have carried out the structural analysis of caffeine-enriched whole coffee cherry extracts, CEWCCE by using 1H and 13C NMR spectroscopy. The changes in 1H chemical shift data in NMR spectra of CEWCCE compared to pure caffeine indicated the formation of complexes between caffeine and chlorogenic acids 13 in aqueous solution. It is based on a rather classical methodology for NMR work The experimental and discussion sections were carried out carefully. For instance, the changes in proton chemical shifts are carefully interpreted taking into account other reasons that can lead to such changes. Therefore I consider it is valuable for publication in Molecules. However the following addressed points should be answered or completed:

Indicate the proton frequency on the figures with proton NMR data (idem for 13C).

  • We have updated this as 1H NMR spectra of A) CEWCCE compared to the spectrum B) GCBE (99% Caffeine) in phosphate buffered D2O pD 7.1 operating at 299.99 MHz, Three methyl groups (H11”, H10”, H12”) with chemical shifts of 3.270, 3.442, and 3.900 ppm and another peak 7.876 ppm for aromatic protons (H8”)

13C NMR spectra of CEWCCE in D2O operating 75.44 MHz.

I not sure that four decimals are required for proton chemical shifts

  • That should be OK

In Figure 1 the meaning of Q should be recalled

  • We have updated these

Line 112 it is written “signals (H12’’) is a doublet” the origin of this pattern should be given.

  • We have given the reference

In Figure 2, to help the reading, the assignments of the NMR signals should be indicated on the different spectra

  • We have updated this as We have updated these as 1H NMR spectra of A) CEWCCE compared to the spectrum of B) GCBE (99% Caffeine) in phosphate buffered D2O pD 7.1 operating at 299.99 MHz (Red: GCBE, Blue: CEWCCE). Dotted lines added for clarity of relative complexation shifts. Three methyl groups (H11”, H10”, H12”) with chemical shifts of 3.270, 3.442, and 3.900 ppm and another peak 7.876 ppm for aromatic protons (H8”)

 In the caption of Figure 3 please insert H8''  after “Aromatic caffeine signal”

  • We have updated this

 In Figure 6, please try to improve the inset: font should be bigger and the black contour may be removed

  • We have updated accordingly

 In Figure 9, font used for the NMR assignments should be bigger (same remarks for Figure 10)

  • We have updated

 In Figure 12, the inset should be replaced by two arrows and the corresponding explanation should be inserted in the figure caption.

  • We have updated this.